# Transformers with Competitive Ensembles of Independent Mechanisms

## Abstract

An important development in deep learning from the earliest MLPs has been a move towards architectures with structural inductive biases which enable the model to keep distinct sources of information and routes of processing well-separated. This structure is linked to the notion of independent mechanisms from the causality literature, in which a mechanism is able to retain the same processing as irrelevant aspects of the world are changed. For example, convnets enable separation over positions, while attention-based architectures (especially Transformers) learn which combination of positions to process dynamically. In this work we explore a way in which the Transformer architecture is deficient: it represents each position with a large monolithic hidden representation and a single set of parameters which are applied over the entire hidden representation. This potentially throws unrelated sources of information together, and limits the Transformer's ability to capture independent mechanisms. To address this, we propose Transformers with Independent Mechanisms (TIM), a new Transformer layer which divides the hidden representation and parameters into multiple mechanisms, which only exchange information through attention. Additionally, we propose a competition mechanism which encourages these mechanisms to specialize over time steps, and thus be more independent. We study TIM on a large-scale BERT model, on the Image Transformer, and on speech enhancement and find evidence for semantically meaningful specialization as well as improved performance.

## 1 Introduction

A major theme throughout the history of deep learning has been the introduction of inductive biases in neural architectures, more recently with a focus on the ability to dynamically keep distinct types of information separated. While an MLP architecture has one large hidden representation at each layer, a convnet keeps different spatial positions' representations separated by default. This separation enables more appropriate reuse of parameters, improving generalization (e.g. compared with a fully connected MLP) by ensuring that some parts of the hidden representation capturing some aspects of the data can remain unchanged when other aspects are changed. Additionally, it is important to be able to reuse parameters in all situations where the parameters are relevant, and not use parameters in positions where they are irrelevant, and this is where attention mechanisms can be very useful.

While dividing information between different positions (for example time steps or spatial positions) is already very useful, it has been recognized from the earliest deep learning work on the notion of disentangling (Bengio, 2009; Glorot et al., 2011; Rifai et al., 2012; Mathieu et al., 2016; Achille & Soatto, 2018) that other features of the data could advantageously be kept well-separated, even over overlapping sets of positions. This has suggested the idea that a model can be decomposed into multiple components, which are often called modules, each operating on a different set of features. Modularity has been identified as an essential ingredient for generalization in machine learning (Ronco et al., 1997; Alet et al., 2018; Goyal et al., 2019). The motivating intuition is that if the relationship between the modules changes between training and evaluation, then a model which keeps these modules sufficiently separate but can adapt how they are combined could be more robust. It can even be robust to changes where the overall data distribution differs between training and evaluation. This has been studied in the causality literature through the notion of "Independent Mechanisms"

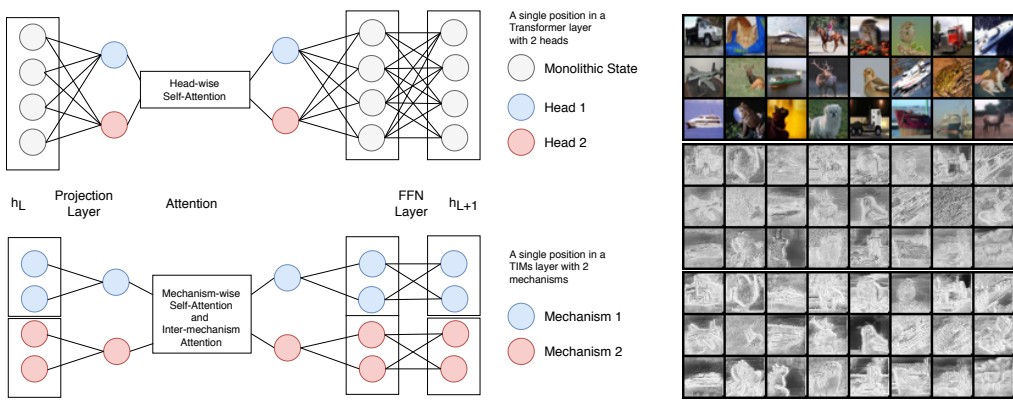

Figure 1: We show a simplified version of the model at a single position to illustrate the difference between heads and mechanisms (left). Heads allow for parallel attention, but the differentiation between heads is transient: it begins with the projection layer and ends immediately following the attention. As a result, most of the parameters are not head-specific. With independent mechanisms, the information is kept well-separated throughout the entire layer, and all of the layer's parameters are specific to a single mechanism. Competition patterns of an unsupervised image Transformer on CIFAR images (top right) with 2 mechanisms shows that mechanisms learn to specialize over foreground and background patterns on an early layer (center right) and become more confident in a later layer (bottom right).

(Peters et al., 2018; Parascandolo et al., 2018) or causal modules, which can be flexibly re-combined, re-used, and re-purposed.

While modularity and independent mechanisms ideas are closely related, the latter has a special focus on the notion that mechanisms should have the ability to remain unchanged when unrelated aspects of the world are changed. In that sense it is a more specific idea which builds on the more general concept of modularity. While the study of independent mechanisms in the context of deep architectures is relatively recent (Goyal et al., 2019; Mittal et al., 2020), a few ideas are considered central. One is that mechanisms are separately parameterized (or dynamically parameterized, with the possibility of separation), which means that the function computed by a module remains the same even as other mechanisms need to be changed. Another central idea is specialization between mechanisms, which is the idea that mechanisms should seek to only model some parts of the world. One way to help accomplish this is by forcing the mechanisms to compete to explain different positions (in time or space), such that some mechanisms would not be used by the model on positions where they are less relevant.

In this work we explore how the idea of independent mechanisms can be beneficial in the Transformer architecture. Transformers (Vaswani et al., 2017) are based on information sharing across positions controlled dynamically by a soft-attention mechanism (Bahdanau et al., 2014), while still using a fully-connected MLP to process the extracted feature vectors (concatenated over a set of attention heads) at each position. An important way in which this improves over convnets is that if this attention becomes sufficiently sparse, then it gains the ability to keep information well-separated between different positions. At the same time, at each position, the Transformer stores a single monolithic hidden representation, over which it applies its entire set of parameters. For example, if we consider a generative model of images of animals in a field, then some of the parameters like those describing how animals have symmetric eyes or a certain number of feet, are only relevant for the positions in the image where the animal is present. A normal Transformer, however, would apply the same parameters to the entire hidden representation at all spatial positions. Additionally, if sources of information need to be accessed over multiple positions, it has no way to keep that information well-separated between parts of the hidden representation, unless a large fraction of the parameters are set to exactly zero. In practice, models tend not to learn these sorts of highly sparse parameter matrices as it is not necessary in order to fit the training set. Thus different underlying factors tend to be freely blended together rather than disentangled: we hypothesize and show empirically that this leads to deteriorated generalization when something about some of these factors changes.

Our newly proposed technique, which we call Transformers with Competitive Independent Mechanisms (TIM) seeks to address this limitation of the Transformer by dividing the hidden representation and parameters into multiple distinct mechanisms. These mechanisms perform self-attention (over input elements) separately, and information is exchanged sparingly between the mechanisms using

attention. Thus the model is naturally compelled to keep multiple information signals well-separated, even within a single position. Moreover, only the parameters corresponding to an activated mechanism are called upon, focusing on one aspect of the hidden representation. The process of selectively activating some mechanisms and not others relies on competition between mechanisms, just like in recurrent independent mechanism (RIMs) (Goyal et al., 2019). We hypothesize and show empirically that this provides an inductive bias encouraging the mechanisms to be more independent and specialized, more robust to changes only affecting other mechanisms.

## 2 TRANSFORMERS WITH COMPETITIVE INDEPENDENT MECHANISMS

### 2.1 PRELIMINARIES

***Multihead Self-attention sub-layer*** The attention mechanism can be formulated as querying a dictionary with key-value pairs (Bahdanau et al., 2014; Vaswani et al., 2017), e.g., $\text{Attention}(Q, K, V) = \text{softmax}(QK^T/\sqrt{d_{model}}) \cdot V$, where $d_{model}$ is the dimensionality of the hidden representations and $Q$ (Query), $K$ (Key), $V$ (Value) are specified as the hidden representations of the previous layer in the so-called *self-attention* sub-layers in the Transformer architecture. The multi-head variant of attention allows the model to jointly attend to information from different representation subspaces, and is defined as $\text{Multihead}(Q, K, V) = \text{Concat}(\text{head}_1, \cdots, \text{head}_H)W^O$, with the heads defined as: $\text{head}_k = \text{Attention}(QW_k^Q, KW_k^K, VW_k^V)$ where $W_k^Q \in \mathbb{R}^{d_{model} \times d_K}, W_k^K \in \mathbb{R}^{d_{model} \times d_K}, W_k^V \in \mathbb{R}^{d_{model} \times d_V}$, and $W^O \in \mathbb{R}^{Hd_V \times d_{model}}$ are project parameter matrices, $H$ is the number of heads, and $d_K$ and $d_V$ are the dimensionalities of Key and Value.

***Group Linear Layer:*** It takes multiple hidden representations, and applies a separately parameterized linear transformation to each. This operation can be efficiently implemented using batched-matrix multiplications. We set the numbers of groups $n_s$ and define a weight tensor $W \in \mathbb{R}^{n_s \times d_{in} \times d_{out}}$. If the input h is shaped as $h \in \mathbb{R}^{n_s \times d_{in}}$, then we can define the layer as: $GroupLinear(h, W, n_s) = [h_j W_j]_{j=1}^{n_s}$

### 2.2 TIM ALGORITHM

We first lay out the parts of a TIM layer and then give more detailed steps in Algorithm 1. We then give a high-level detail of how to turn a transformer layer into a TIM layer in a typical implementation (Section 2.3). An illustration of how independent mechanisms differ from heads is given in Figure 1.

#### 2.2.1 COMPETITION BETWEEN DIFFERENT MECHANISMS

Aside from having separate parameters and only exchanging information via inter-mechanism attention, we wanted to create a stronger inductive bias to encourage the mechanisms to specialize. To do this, we created a competition system in which each mechanism has a layer which outputs a single scalar value (as a function of the current layer's representation), and these are passed through a softmax over the different mechanisms (this softmax is applied position-wise and separately for each layer). The value of this softmax is then used to weight how much each mechanism is allowed to update its representation after the self-attention. This competition score is computed as $c = \text{softmax}(\text{GroupLinear}(h, W^c, n_s))$, where we note that each mechanism has its own parameters for the layer (hence the use of a Group Linear layer instead of a normal linear layer). Thus the $n_s$ modules have a per-step weighting for how much they are able to read during the later self-attention stage. Thus if one mechanism wants to perform attention on a given position, it suppresses the other mechanisms on that position. We found that this often improved results and that these softmax scores are fairly interpretable as a measure of specialization. Exact equations for this step are given in Step 1 and used in Step 2 in Algorithm 1 in the appendix.

#### 2.2.2 EACH MECHANISM SHARES INFORMATION ACROSS TIME AND PROCESSES INFORMATION

This step allows each mechanism to have its own independent dynamics, which are themselves similar to a normal transformer layer. These independent dynamics allow each mechanism to read information from other time steps using attention and process that information using FFN layers. We modify the self-attention sub-layer and feed-forward sub-layers (FFN) to be mechanism-wise as well

as position-wise, with separate parameters for each mechanism. Additionally, the layer-normalization is modified to be performed separately for each mechanism. The projections and FFN sub-layers can be modified by replacing the linear layers with group linear layers. When performing the self-attention itself, the mechanisms behave the same as heads, and thus we can use the same type of multi-head attention process, so long as the total number of heads is divisible by the number of mechanisms. One notable property is if TIMs only consisted of this part of the model (independent dynamics) by itself, then each TIM would be a completely independent transformer model with its own forward pass and its own parameters. Steps 2 and 4 in the appendix, Algorithm 1 give more detail on this step.

### 2.2.3 Attention is used to Communication Information Between Different Mechanisms

Although we allow each TIM to remain independent and process information independently, it is also important to allow the different mechanisms in TIMs to share information between each other (in case the TIMs are not truly fully independent). To do this we use a standard multi-head attention sub-layer to share information between the mechanisms, which is done in a position-wise fashion. We made this attention mechanism relatively small, with just 2 heads with 32 units each. This is because we want the different mechanisms to be as independent as possible, and thus only share small amounts of high level information. This can be thought of as another attention layer, where we treat the different mechanisms as positions, and perform this attention in parallel over the different steps in the sequence. More details on this are given in Step 3 in the appendix's Algorithm 1.

### 2.3 Implementing and Integrating TIM

The TIM layer is a drop-in replacement for a standard Transformer layer and turning an existing Transformer layer into a TIM layer is surprisingly straightforward. It is a drop-in replacement for a single layer which can be flexibly used in a variety of models and architectures (both encoders and decoders). A simple strategy is if a normal hidden representation is of shape $(T, b, d_{model})$, then our TIM hidden representation should be reshape-able to $(T, b, n_s, d_{model}/n_s)$. First, each layer linear layer in the existing Transformer layer should be replaced by a group-wise linear layer implemented using batch matrix multiplication. Second, so long as the number of heads is divisible by the number of mechanisms, the self-attention does not need to be changed, since mechanisms behave interchangeably with heads in this part of the model. Third, the inter-mechanism communication can be added as a drop-in module into the Transformer layer. Finally, the competition layer is just a single layer with a softmax, which can easily be added.

Although TIM is a drop-in replacement for a normal Transformer layer, there are a few subtleties that must be considered for successful integration. First, if the total size of the hidden representation is kept the same, integrating TIM drastically reduces the total number of parameters in the model because all of the linear layers are replaced by grouped-linear layers (which can be thought of as having a block-sparse structure). This step by itself reduces the number of parameters by a factor of $n_s$, but a TIM layer also adds new parameters to the model through the addition of the Inter-mechanism Attention Sub-Layer and Mechanism-Competition Sub-Layers, although both of these are rather small. In practice a TIM layer usually reduces the number of parameters by about 30-40%, depending on the exact hyperparameters. To compensate for this, in all of our experiments we increased the total hidden size to match the number of parameters of the original model, usually by about 20%.

Additionally, while we initially thought that it would make sense to replace every Transformer layer with a TIM layer, when we analyzed the mechanism-competition, we found that it was almost always completely flat on the first layer, which suggested to us that the first two layers as well as the last layer should be kept as normal Transformer layers.

## 3 Related Work

***Specialization and Competition over heads in Transformers.*** Cui et al. (2019) proposed a mixed multi-head attention mechanism which forces some heads to learn specific patterns, such as attending to precedent/local tokens only. Clark et al. (2019) studied which positions attention heads focus on and found that some heads have specific patterns, such as attending locally. Vig et al. (2020) showed that the heads in a model of protein sequences are semantically meaningful. An et al. (2020)

considered adding a repulsive force to the heads in Transformers to try to make them more specialized. In our view, this evidence for specialization over heads is complementary with our results.

***Independent Mechanisms and Modularity in Transformers.*** We're not aware of any work which breaks a Transformer's hidden representation into multiple mechanisms with separate parameters which interact through attention, though some works hint at this direction. The Group Transformer (Park et al., 2020) replaces the fully-connected layers with group-linear layers and uses low-rank layers to pass information between the groups. The universal transformer (Dehghani et al., 2018) shared parameters between layers and updated using gating, and this gating could behave similarly to the competition that we propose but lacks the idea of having multiple mechanisms.

***Independent Mechanisms in Recurrent Networks.*** The idea of independent mechanisms has seen a significant amount of focus in recurrent networks (Goyal et al., 2019). The idea is to parameterize the model as an ensemble of mechanisms, having their own dynamics, but sparingly interacting with each other using a bottleneck of attention. In the case of recurrent networks, dividing the hidden representation into mechanisms has the advantage that at a particular time-step, only a fraction of mechanisms can be active, and hence computation is sparse in time, where in the case of transformers, imposing the idea of independent mechanisms in some higher layers has the added advantage that computation can be sparse both in space (i.e., position ) as well as time.

## 4 EXPERIMENTS

We seek to answer two questions in our experiments. First, do the mechanisms that we learn with TIM specialize in sensible and semantically meaningful ways? We analyze this both on toy datasets where we have clearly independent mechanisms by construction (Figure 2) and on large-scale realistic speech and NLP tasks (Figure 3 and Figure 4). Our second question is how using a model which learns these independent mechanisms leads to better quantitative performance, both on the original task and on transfer learning, which we demonstrate in Figure 1 and Table 2.

### 4.1 IMAGE TRANSFORMER: EVIDENCE OF SPECIALIZATION

We integrated TIM into the Image Transformer, which is a generative model which generates an image pixel-by-pixel, with a small-scale variant of the GPT-2 architecture (Karpathy, 2020; Radford et al., 2019). We first considered a pedagogic task in which the dataset consists of two clearly independent mechanisms. Our synthetic task uses MNIST digits (LeCun & Burges, 1998) and CIFAR images Krizhevsky (2009) of small realistic images of animals and vehicles. Each example in our constructed dataset consists of an MNIST digit on its left-side and a CIFAR image on its right-side, with these two examples selected randomly. It is clear that two sides of the image are independent and have completely different types of content, and thus it is natural for each mechanism to specialize over a single side.

When training with TIM on this dataset, we found that we were able to nearly exactly recover a competition pattern in which the mechanisms specialize over the two sides of the image (Fig. 2, middle). Intriguingly, this specialization does not appear at the very beginning of training, in which the mechanisms mostly specialize over the lightness or darkness of the pixels. However as training progresses, the two sides of the image become increasingly specialized to one mechanism or the other (Figure 2). We also experimented with the CIFAR-10 dataset, and found that integrating TIM led to superior test-set likelihoods. Moreover we visualized the competition pattern with TIM on CIFAR-10 and found a specialization between foreground and background regions in the images (Fig. 1, right).

### 4.2 SPEECH ENHANCEMENT

Speech enhancement aims to improve the quality of speech recordings. A speech signal captured in real environments, in fact, is often corrupted by noise and reverberation that might severely affect its intelligibility. Speech enhancement has long been studied in the research community (Jacob Benesty & Chen, 2015). Traditional approaches were based on signal processing techniques such as spectral-subtraction or Wiener filtering (Boll, 1979; Ephraim & Malah, 1984; Scalart & Filho, 1996). The idea behind these methods is to estimate the noise in non-speech segments and remove it from speech regions. End-to-end deep learning-based speech enhancement has turned out to significantly outperform traditional signal processing methods, and recently using Transformers

Figure 2: We trained an Image Transformer (pixel-by-pixel, raster-order generative model) on a dataset where we construct new images in which the left-half of each image is an MNIST digit and the right-half of the image is a random CIFAR example (left). We found that with $n_s = 2$, the two mechanisms in TIM learn to specialize over the two sides of the image, with one TIM only activating on the MNIST digit and one TIM only activating on the CIFAR example (activation patterns shown in center). We measured the relative variance in the mechanism-activation over the two sides of the image, showing that the specialization becomes much stronger over the course of training (right).

has led to promising performance (Kim et al., 2020). We believe that TIM fits well with this task because the traditional technique of decomposing the signal into speech and noisy parts and then analyzing these two signals separately embodies the desiderata of independent mechanisms.

Table 1 (left) compares the performance achieved by TIM with other recent systems on the widely-studied Deep Noise Suppression (DNS) dataset (Reddy et al., 2020). DNS is a large corpus composed of roughly 441 hours of clean and noisy speech samples. The clean speech is artificially corrupted with noise sequences from the audioset database, which contains two million human-labeled clips drawn from YouTube videos and belong to about 600 audio events. Noise is added to the clean speech signal using a random signal-to-noise-ratio (SNR) ranging from 0 to 40 dB. We replaced all Transformer layers except for the first two and the last layer with TIM layers and we increased the total number of hidden units and heads (by about 20%) to match the number of parameters of the baseline, and we used two mechanisms (but achieved slightly worse yet better-than-baseline results with $n_s = 4$). The systems are evaluated with the Perceptual Evaluation of Speech Quality (PESQ) score (Rix et al., 2001). To assess the generalization capability of TIM, we tested our model on the Voicebank test-set as well (see Table 1-right). Voicebank (Thiemann et al., 2013), in fact, is characterized by noisy conditions different from that of the DNS dataset used for training.

The results, shown in Table 1, highlight that TIM slightly outperforms the recently-proposed PocoNet (Hu et al., 2020) model, which uses additional data and has 8 times the parameters of TIM. To the best of our knowledge, TIM achieves the best PESQ performance so far published in the literature on the DNS dataset. Qualitatively, we found that the competition scores matches our intuition. Indeed, the two mechanisms clearly specialize over speech and non-speech parts of the audio sequence, as shown in Figure 3. Moreover, we intriguingly found that this competition between mechanisms is consistent across layers, starts out with low confidence, and becomes increasingly confident in later layers. Compared to a standard Transformer, TIM shows superior generalization capabilities. This interesting feature can be appreciated in Table 1 (right), where we tested our model on a different dataset (VoiceBank). In mismatch conditions, the competition mechanism seems to play a crucial role. This finding agrees with our intuition, according to which employing specialized and competing modules can make the model less affected by irrelevant changes of the input distribution.

### 4.3 BERT PRE-TRAINING AND FINE-TUNING

BERT (Devlin et al., 2018) is one of the most popularly used methods to learn the representation of natural language. The BERT model uses a multi-layer Transformer encoder and is trained by the masked language modeling task using Web data corpus (Liu et al., 2019). The pre-trained contextual sentence representations have been shown to be effective in a large number of downstream tasks.

For BERT, we replaced all of the transformer layers except for the first two layers and the last layer with TIM layers (we also report a result where all layers are TIM layers, showing that it leads to worse performance). We used two mechanisms and evenly increased the number of hidden units and total number of heads across all layers to match the number of parameters in the baseline model.

**Pre-training** Following Devlin et al. (2018), we used English Wikipedia corpus and BookCorpus for pre-training. By concatenating these two datasets, we obtained a corpus with roughly 3.4 billion

Table 1: We trained TIM on the DNS speech enhancement dataset and evaluate on the DNS test-set (left). To assess zero-shot transfer generalization we evaluate it on the voicebank test-set as well (right). The performance is reported on wideband PESQ (higher is better). TIM reaches state-of-the-art performance and shows improved generalization capabilities in mismatch conditions. Results with (*) used additional outside datasets for training. For external baselines (a) is (Choi et al., 2020), (b) is (Koyama et al., 2020), and (c) is (Isik et al., 2020).

| Models (Trained on DNS) | Param (M) | DNS (PESQ) |
|---|---|---|
| Noisy - no reverb | n/a | 1.582 |
| U-Net-MultiScale+ (a) | 3.5 | 2.710 |
| Conv-TasNet (b) | 5.1 | 2.730 |
| PoCoNet (c) | 50.0 | 2.722 |
| PoCoNet-SSL* (c) | 50.0 | 2.748 |
| Transformer Baseline | 6.1 | 2.727 |
| TIM-NoComp($n_s$= 2) | 6.0 | 2.754 |
| TIM-Comp($n_s$= 2) | 6.0 | 2.742 |
| TIM-Comp($n_s$= 4) | 6.0 | 2.730 |

| Models (Trained on DNS) | VoiceBank (PESQ) |
|---|---|
| Noisy - no reverb | 1.970 |
| Transformer Baseline | 2.517 |
| TIM-NoComp($n_s$=4) | 2.503 |
| TIM-Comp($n_s$=2) | 2.575 |
| TIM-Comp($n_s$=4) | 2.540 |

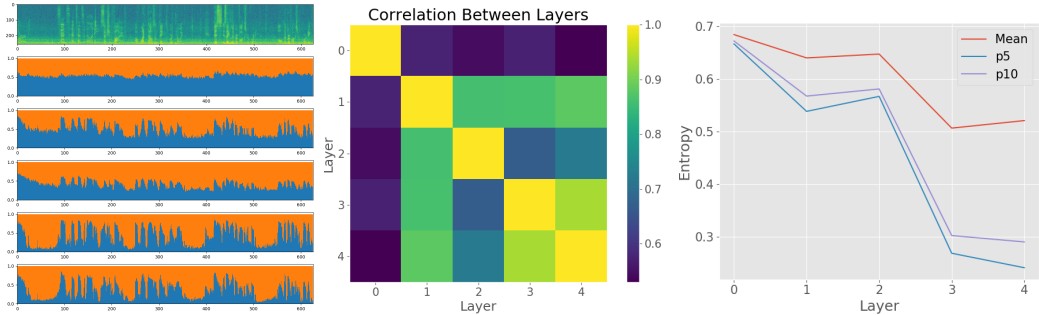

Figure 3: An examples of a speech signal (left) with their respective competition patterns over five successive TIM layers (ordered from top to bottom). In the early layers, the competition is uncertain, but becomes more certain in the deeper layers. This is further quantified in a correlation matrix of competition over layers (middle) and a plot showing that competition entropy drops in later layers, especially at the lowest percentiles (right).

words in total. We trained all model variants with the same procedure and hyperparameters which were tuned on the BERT baseline model. All models were run on 16 NVIDIA Tesla V100 GPUs.

**Fine-tuning** We used MNLI, MNLI-MM, QNLI, SST-2 and STS-B from the GLUE (**G**eneral **L**anguage **U**nderstanding **E**valuation) dataset (Wang et al., 2018) as the downstream tasks to evaluate the performance of the pre-trained models. Ideally the features learned by BERT would remain useful on these distinct tasks which have relatively small training sets.

**Results** The overall comparison results are shown in Table 2. We found that both TIM-NoComp and TIM-Comp achieve lower perplexities (masked language modeling loss) on the validation dataset compared to the two BERT baselines. We found generally better and more reliable (less variance between seeds) results when fine-tuning experiments with TIM. These empirical results show that our proposed TIM is a better model architecture in a wide range of natural language applications.

## 4.4 DISCUSSION: RNN MODULARITY VS. TRANSFORMER MODULARITY

A single layer RNN is already a fairly powerful model which can have strong priors to inform how to select mechanisms (or modules more generally). However, a single layer Transformer is a rather weak model, as it can only base its representations on a single round of attention based upon the individual tokens and their position encoding. We've consistently found that the quality of the mechanism-competition is poor in the first layer of a Transformer network and that performance is substantially improved by making the early layers use ordinary Transformer layers rather than TIM layers. This is in contrast with what has been observed with RNNs, where improvements can be obtained by using multiple modules or multiple mechanisms even in a single layer model.

Table 2: We compare the baseline BERT models to TIM with and without competition, on both validation likelihood (perplexity) and NLP fine-tuning tasks (reported as accuracy, with median and standard deviation over five fine-tuning trials with different seeds). We also show that it is essential to make the first and last layers normal Transformer layers and not TIM layers (TIM-All-Layers).

| Result | BERT | BERT-130M | TIM-All-Layers | TIM-NoComp | TIM-Comp |
|---|---|---|---|---|---|
| TIM Layers | 0/12 | 0/12 | 12/12 | 9/12 | 9/12 |
| Parameters | 110M | 130M | 110M | 130M | 130M |
| Competition? | ✗ | ✗ | ✗ | ✗ | ✓ |
| Valid-NLL | 2.096 | 2.040 | 2.112 | 2.033 | 2.027 |
| MNLI-M | $84.93 \pm 0.15$ | $85.37 \pm 0.29$ | $84.19 \pm 0.34$ | $85.89 \pm 0.17$ | $85.28 \pm 0.22$ |
| MNLI-MM | $84.91 \pm 0.18$ | $85.28 \pm 0.27$ | $84.55 \pm 0.15$ | $85.80 \pm 0.07$ | $85.17 \pm 0.18$ |
| QNLI | $91.34 \pm 0.21$ | $91.84 \pm 0.32$ | $91.37 \pm 0.59$ | $91.78 \pm 0.14$ | $91.97 \pm 0.20$ |
| SST-2 | $92.88 \pm 0.33$ | $92.75 \pm 0.26$ | $92.52 \pm 0.56$ | $92.75 \pm 0.13$ | $92.97 \pm 0.25$ |
| STS-B | $89.43 \pm 0.25$ | $89.34 \pm 0.15$ | $88.20 \pm 0.32$ | $88.52 \pm 0.28$ | $89.63 \pm 0.05$ |

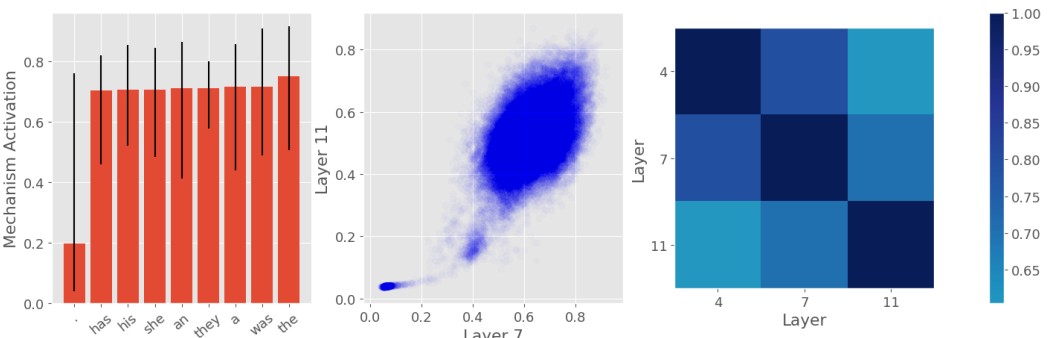

Figure 4: On a BERT model, we show the minimum, average, and maximum mechanism competition values of some selected common tokens (left). One mechanism clearly specializes over the period between sentences, yet the high difference between the minimum and maximum values suggest that these differences are contextual and not a static function of the token. In particular we found that the modular activation for a period depends on whether it is used to mark the end of a sentence or whether it is used as part of a number or a URL. Moreover, the mechanism activation is highly correlated between layers (scatter-plot in center, correlation matrix on right).

## 5 CONCLUSION

Scaling to extremely large Transformers with a very large number of hidden units for each position has become one of the dominant paradigms in applied machine learning. This work explores a new direction in the structure of the Transformer architecture which will become increasingly important as models become larger and researchers seek to model more complex phenomena. Evidence suggests that the Transformer's success is a result of its use of attention to communicate information between positions, which allows for effective and precise transmission of information even over very long sequences (Kaplan et al., 2020). At the same time, each position within a Transformer is still represented with a single monolithic hidden representation, and a set of parameters which is applied over the entire hidden representation. Our newly proposed technique, TIM, has shown that it is possible to make the Transformer even more dynamic by breaking the hidden representation and layers into multiple mechanisms which interact via attention and have an inductive bias towards specialization. We show that these mechanisms specialize over distinct parts of the data and improve results across diverse types of data. These results suggest that there is room to improve the structural inductive biases in the Transformer and point towards an increasingly central area of future research as state-of-the-art Transformers, and the tasks they're trained on, become larger and more diverse.

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

# A  EXPERIMENT DETAILS

## A.1  IMAGE TRANSFORMER DETAILS

For the Image Transformer, we used a baseline with 6 Transformer layers. As in the other experiments, we made the first 2 layers use TIMs as well as the last layer. We ran each experiment for 30 epochs with a batch size of 24, and otherwise used the same training hyperparameters as the minGPT repository (Karpathy, 2020). We used the Adam optimizer with warmup, with Betas = (0.9, 0.95). Our baseline model had 8 heads (total per-layer) and a layer hidden size of 184. When using TIMs, we increased this (for all layers) to 10 heads and a hidden size of 200. This led to the baseline and the TIM model having roughly the same number of total parameters.

## A.2  SPEECH ENHANCEMENT DETAILS

**Datasets**  Neural speech enhancement systems are trained using a parallel corpus of noise and clean examples, which are generated by artificially contaminating clean speech with disturbances such as additive noise and reverberation (Ravanelli & Omologo, 2015). The speech enhancement models considered in this work are trained with the DNS dataset (noisy, no reverb) (Reddy et al., 2020), which is a synthetic corpus recently made publicly available by Microsoft. This corpus is extremely suitable for our study because it is quite big (441 hours) and contains a large variety of possible noises (from 600 different categories). To the best of our knowledge, it is the biggest open-source speech enhancement dataset. Moreover, it has been the object of an international challenge on speech enhancement[1]. This gave us the possibility to compare our TIM with the best systems submitted to this competition.

For evaluation, we used the test sets of the DNS and Voicebank datasets (Thiemann et al., 2013). The latter has been adopted to study a transfer learning scenario, where different datasets are used for training and evaluation purposes. Voicebank, in fact, is generated with noisy sequences different from the one contained in the DNS corpus. Since Voicebank is released at 48 kHz, the original raw waveforms were downsampled from 48kHz to 16kHz.

**Model Architecture**  The proposed TIM is fed with noisy speech and estimates clean speech at the output. More precisely, we estimate the log-spectral magnitude of the clean signal. Mean Squared Error (MSE) between the clean and the corresponding noisy signal is used as cost function. The input waveform is transformed with the Short-Term Fourier Transform (STFT) based on 512 frequency points and window length of 32 ms with 16 ms overlap.

Before adding the transformer layers, we employ four 1D convolutional layers that act as a pre-encoder module. This is done to replace positional encoding from the original transformers and inject relative location information to the frames in the sequence (Kim et al., 2020; Fu et al., 2020). The four convolutional layers are based on 1024, 512,128, and 256 channels, respectively. The kernel size is 3. After the convolution, each layer applies layernorm followed by LeakyReLU. The Transformer part is composed of 8 encoder blocks with a hidden size of 512. In order to employ approximately the same number of parameters (i.e, 6 million), the baseline transformers used a hidden size of 256. We used 16 attention heads, a dropout rate of 0.1, and LeakyReLU activations. We kept the number of heads the same as in the baseline model. To follow the real-time processing restriction in DNS challenge, a causal setting is adopted to all our models with access to 32 ms of future frames. Attention masks are also applied to the self-attention layers to prevent using the future information.

**Training**  We followed the exact same training procedure for the baseline model and the TIMs model, with both trained for 50 epochs. We used the standard variant of the Adam optimizer with a batch size of 16. The initial learning rate was set to 0.0002 and halved when the validation score decreased for 5 epochs. We reported test set performance at the epoch with the best validation score, which in practice was near the end of training. Both models train for about 50 hours on a single Nvidia V100 GPU.

## A.3  BERT PRE-TRAINING AND FINE-TUNING DETAILS

BERT (Devlin et al., 2018) is one of the most popularly used methods to learn the representation of natural language. The BERT model uses a multi-layer Transformer encoder and is trained by the masked language modeling task using Web data corpus (Liu et al., 2019). The pre-trained contextual sentence representations have been shown to be effective in a large number of downstream tasks.

To validate our proposed architecture, we conduct experiments to compare TIM with Transformer on the language pre-training task. For our model, we replace all of the transformer layers except for the first two layers and the last layer with TIM layers (we also report a result where all layers are TIM layers, showing that it leads to worse performance). We scaled the dimensionality of the hidden nodes and the inner-layer of the FFN sub-layer

---

[1] `https://dns-challenge.azurewebsites.net/Interspeech2020`

are set to, the number of mechanisms is set to 2 and the number of heads is set to 16. We mainly test two TIM variants, TIM without competition (TIM-NoComp) and TIM with competition (TIM-Comp).

For a fair comparison, we set one baseline as a 12-layer Transformer with 130M parameters (BERT-130M). The size of hidden nodes and the inner-layer of the FFN sub-layer are set to 768/4096, and the number of heads is set to 12. We also use the standard BERT-Base model (110M parameters) as another baseline.

**Dataset**  Following Devlin et al. (2018), we use English Wikipedia corpus[2] and BookCorpus[3] for pre-training. By concatenating these two datasets, we obtain a corpus with roughly 3400M words in total. We follow a couple of consecutive pre-processing steps: segmenting documents into sentences by Spacy [4], normalizing, lower-casing, and tokenizing the texts by Moses decoder (Koehn et al., 2007), and finally, applying *byte pair encoding* (BPE) (Sennrich et al., 2016) with setting the vocabulary size $|V|$ as 32,678.

**Optimization**  Following the standard settings used in many previous works Devlin et al. (2018); Liu et al. (2019), we train the models for $1000k$ steps with setting the batch size as 256 and the maximum sequence length as 512. For all the models to compare, we set the masked probability $p$ to be 0.15. We follow previous works to replace 80% of the masked positions by [MASK], 10% by randomly sampled words, and keep the remaining positions unchanged. We choose the most widely used Adam Kingma & Ba (2014) as the optimizer, and set the hyper-parameter $\beta$ as $(0.9, 0.98)$. The learning rate is set as 1e-4 with a $10k$-step warm-up stage and then decays linearly to zero. We set the dropout probability as 0.1. All models are run on 8 NVIDIA Tesla V100 GPUs.

**Fine-tuning**  We use the GLUE (**G**eneral **L**anguage **U**nderstanding **E**valuation) dataset (Wang et al., 2018) as the downstream tasks to evaluate the performance of the pre-trained models. Reporting large-scale task performance or averaged performance over all tasks depends on our choice. Same to the pre-training, we use Adam as the optimizer and set the hyper-parameter $\beta$ as $(0.9, 0.98)$. Following all previous works, we apply the hyper-parameter search (over $\beta$ and learning rate) during the fine-tuning for each downstream task. Each configuration was run for five times with different random seeds, and the median and standard deviation over these five results on the development set was be used as the performance of one configuration.

**Results**  The overall comparison results are shown in Table 2. It is easy to find that both TIM-NoComp and TIM-Comp achieve lower perplexities (masked language modeling loss) on the validation dataset compared to the two BERT baselines. On the downstream tasks, the two TIM variants are also slightly better than the BERTs on all tasks. Those empirical results show that our proposed TIM is a better model architecture in a wide range of natural language applications.

Similar to previous analysis, we further study the competition patterns in the TIM-Comp model to investigate how the competitive module behaves.

---

[2]https://dumps.wikimedia.org/enwiki

[3]As the dataset BookCorpus (Zhu et al., 2015) is no longer freely distributed, we follow the suggestions from Devlin et al. (2018) to crawl from smashwords.com and collect BookCorpus by ourselves.

[4]https://spacy.io

# B    DETAILED ALGORITHM DESCRIPTION

---

**Algorithm 1** A single TIM Encoder-Layer

---

*Hyperparameters:* Number of mechanisms $n_s$, key size $d_k$, value size $d_v$, number of heads for self-attention $H$, number of heads for inter-mechanism attention $H_c$. We set $d_{mech} = d_{model}/n_s$ and $d_{ffn-m} = d_{ffn}/n_s$

*Input:* An input hidden representation $h$ for a single example of shape $(T, bs, d_{model})$.

***Step 1: Compute Mechanism Competition***
    $W^c \in \mathbb{R}^{n_s \times d_{mech} \times 1}$
    $c = \text{softmax}\big(\text{GroupLinear}(h, W^c, n_s)\big)$

***Step 2: Mechanism-wise self-attention sub-layer***
    $W_2^Q, W_2^K \in \mathbb{R}^{n_s \times d_{mech} \times H d_K}$
    $W_2^V \in \mathbb{R}^{n_s \times d_{mech} \times H d_V}$,
    $W_2^O \in \mathbb{R}^{n_s \times H d_V \times d_{mech}}$
    $Q = \text{GroupLinear}(h, W_2^Q, n_s)$
    $K = \text{GroupLinear}(h, W_2^K, n_s)$
    $V = \text{GroupLinear}(h, W_2^V, n_s)$
    $M := \text{Attention}(Q, K, V, n_s H)$
    $M := \text{GroupLinear}(M, W_2^O, n_s)$
    $h := \text{norm}(h + c \odot M, n_s)$

***Step 3: Inter-mechanism Attention Sub-Layer***
    $W_3^Q, W^K \in \mathbb{R}^{n_s \times d_{mech} \times H_c d_K}$
    $W_3^V \in \mathbb{R}^{n_s \times d_{mech} \times H_c d_V}$
    $W_3^O \in \mathbb{R}^{n_s \times H_c d_V \times d_{mech}}$
    $Q = \text{GroupLinear}(h, W_3^Q, n_s)$
    $K = \text{GroupLinear}(h, W_3^K, n_s)$
    $V = \text{GroupLinear}(h, W_3^V, n_s)$
    Reshape Q,K, and V to $(n_s, T * bs, H_c * d)$.
    $M := \text{Attention}(Q, K, V, H_c)$
    Reshape M to $(T, bs, n_s * H_c * d_v)$.
    $M := \text{GroupLinear}(M, W_3^O, n_s)$
    $h := \text{norm}(h + M, n_s)$

***Step 4: Mechanism-wise, Position-Wise, FFN Sub-Layer***
    $W^{(1)} \in \mathbb{R}^{n_s \times d_{mech} \times d_{ffn-m}}$
    $W^{(2)} \in \mathbb{R}^{n_s \times d_{ffn-m} \times d_{mech}}$.
    $\text{F} = \text{GroupLinear}(\sigma(\text{GroupLinear}(h, W^{(1)})), W^{(2)})$
    $h := \text{norm}(h + \text{F}, n_s)$

---

