# OpenReview forum: "Transformers with Competitive Ensembles of Independent Mechanisms"
_ICLR.cc/2021/Conference — Reject_

### Official Review · AnonReviewer4 · 2020-10-27
**Interesting problem statement but experimental setup is not convincing or rigorous**

**Rating:** 4
**Confidence:** 4

**Review:**

Authors tackle the problem of making vanilla Transformer better by handling different positions (in data) via independent and competing mechanisms. Formulation is interesting but experimental setup has various deficiencies:
1. It is quite heuristical to propose to use first two and last layers of Transformer to be vanilla. This suggests that either (1) proposed solution has something deeper going on (uninvestigated), or (2) it is an experimental thing that works for datasets used, or (3) "competition" should not happen in initial and later layers of network.
2. To provide proof-of-concept for independent mechanisms, more complex problems should be picked like image segmentation and speech source separation (more than 2 sources), and they should be investigated with more than 2 mechanisms.
3. Results should also be provided for TIM without matching its number of parameters to vanilla Transformer. This is to see if TIM still gives good performance or not. Also, in my opinion, increasing number of heads of TIM to match parameters with vanilla is not fair. It usually gives performance improvement. IMO only number of hidden layer units may be increased for matching #parameters.
4. DNS performance is good but best is achieved without competition mechanism, which shows that independence and competitiveness are not complementary?
5. Improvement on VoiceBank is decent but baseline chosen in very far away from SOTA. Check: https://arxiv.org/abs/2006.12847

---

### Official Review · AnonReviewer1 · 2020-10-28
**Interesting idea but the results are not convincing**

**Rating:** 5
**Confidence:** 5

**Review:**

This paper proposes an independent mechanism that divides hidden representations and parameters into multiple independent mechanisms. The authors claim that the mechanism benefits the computation of sparse tensors; it does learn better inductive biases than a sizeable monolithic model. This idea is particularly similar to Recurrent Independent Mechanisms (RIM) [1], mentioned in the paper. The main contribution of this work is introducing competition between independent mechanisms. The authors evaluate their models on the image transformer model, speech enhancement, and NLP tasks.

The main thing that has been missing in this paper is the fine-details of the approaches.

I think the paper has exciting ideas on reconstructing the architecture of the Transformer. However, the proposed models only give marginal improvements; thus it is tough to find the reasons for using this model.

To have a stronger claim, I suggest adding a comparison to RIM on Sequential MNIST Resolution Task to show that independent mechanisms can benefit the generalization performance.

Strengths:
- The authors introduce a novel transformer architecture to split the parameters into independent sections for a better inductive bias

Weaknesses:
- The performance improvement is not consistent in all experiments. The competition mechanism benefits only some tasks.
- The evaluation results are not convincing

Several issues and questions for the authors:
1. Can you show the dynamics of the competition between mechanisms over time?
2. In terms of evaluation, I didn't see any baseline for the Image Transformer model. It isn't easy to understand the significance of the proposed method.
3. Please consider citing the relevant papers [2] [3]

***Post-Rebuttal***

> I want to thank the author for addressing my concerns. However, the authors did not address the issues of the evaluation. I think the paper can be further improved by providing more convincing results and analysis. For me, the significance of the proposed method is minimal. Thus, I will not change my score.

References
1. Recurrent Independent Mechanisms https://arxiv.org/pdf/1909.10893.pdf
2. LeCun, Y., Cortes, C., & Burges, C. J. (2010). MNIST handwritten digit database.
3. Krizhevsky, A. (2009). Learning Multiple Layers of Features from Tiny Images.

---

### Official Review · AnonReviewer3 · 2020-10-28
**Simple and efficient Transformer architecture tweak to allow for non-uniform data modelling**

**Rating:** 7
**Confidence:** 4

**Review:**

This paper discusses the potential limitation of Transformers on modelling data with clear non-uniform structure. Authors propose to integrate a useful structural inductive bias which helps to separate the information in hidden states of the Transformer (instead of using a single monolithic hidden representation).

The experiments on a toy image Transformer and speech enhancement demonstrate the effectiveness of the proposed approach. The experiments on BERT pre-training and fine-tuning do not bring much to the table, as the text data is relatively uniform and does not benefit much from the introduced structural inductive bias. As a result, the accuracy of the proposed model on GLUE benchmark is close to the accuracy of BERT baseline.

One obvious limitation of the proposed approach is the necessity to know the number of "independent" components in the data. It would be interesting to see whether it can be generalized to overcome this limitation, e.g., by choosing the number of independent mechanisms automatically.

---

### Official Review · AnonReviewer2 · 2020-11-05
**An interesting idea but a rush paper**

**Rating:** 4
**Confidence:** 4

**Review:**

# Summary of the paper
It is motivated from a perspective that a model should be decomposed into independent modules/mechanisms, such that the model is robust to data distribution shift. A TIM layer is proposed to replace the Transformer encoder layer, where a mechanism competition is introduced to suppress all but one group. Experiments are conducted on different Transformer architectures with a replacement of encoder layers by TIM layers in a) autoregressive image generation with Image Transformer, b) speech enhancement with a customized Transformer and c) BERT on language modeling.

# Pros
The idea that grouping the hidden activations by multiple mechanisms is a promising inductive bias towards out-of-distribution generalization. It is an interesting attempt to integrate this idea to Transformers.

# Cons
1. The presentation is terrible: the method description is basically throwing the code (Algorithm 1) to the audience without much explanation. It is hard to understand the TIM layer if neither diagrams nor equations are supplied.
2. I didn't get what does "mechanism" mean in Algorithm 1. The competition will for sure suppress some activations but how can you guarantee that different groups will learn different behaviors or achieve disentanglement?
3. Experiments only demonstrate n_s = 2 mechanisms. Is n_s > 2 even working? All results suggest that TIM can make some neurons focusing on the foreground, which isn't that surprising as this is most attention based models aiming at. Do you have results showing that TIM can learn diverse functionalities from data which can be interpreted as mechanisms?
4. Figure 1 is confusing: how can we know the difference between "head-wise self-attention" and "Mechanism-wise Self-Attention and Inter-mechanism Attention", which are all represented by a blackbox. What is the competition patterns? How are they computed?
5. Algorithm 1: functions such as GroupLinear(), Attention() are used before defining.
6. Figure 2 (middle): what do you mean "TIM learn to specialize over the two sides of the image"? Also, what are these figures showing?

---

### Decision · Program_Chairs · 2021-01-07
**Final Decision**

**Decision:**

Reject

**Comment:**

The reviewers agree that the idea of introducing structural biases in the attention mechanism is interesting but the results and presentation right now is not convincing. Improvements are seen on only some datasets and the comparisons are not exact.
A reject.